# Structure of the human core transcription-export complex reveals a hub for multivalent interactions

Thomas Pühringer[1†], Ulrich Hohmann[1,2†], Laura Fin[1], Belén Pacheco-Fiallos[1], Ulla Schellhaas[1], Julius Brennecke[2], Clemens Plaschka[1]*

[1]Research Institute of Molecular Pathology (IMP), Vienna BioCenter (VBC), Vienna, Austria; [2]Institute of Molecular Biotechnology of the Austrian Academy of Sciences (IMBA), Vienna, Austria

**Abstract** The export of mRNA from nucleus to cytoplasm requires the conserved and essential transcription and export (TREX) complex (THO–UAP56/DDX39B–ALYREF). TREX selectively binds mRNA maturation marks and licenses mRNA for nuclear export by loading the export factor NXF1–NXT1. How TREX integrates these marks and achieves high selectivity for mature mRNA is poorly understood. Here, we report the cryo-electron microscopy structure of the human THO–UAP56/DDX39B complex at 3.3 Å resolution. The seven-subunit THO–UAP56/DDX39B complex multimerizes into a 28-subunit tetrameric assembly, suggesting that selective recognition of mature mRNA is facilitated by the simultaneous sensing of multiple, spatially distant mRNA regions and maturation marks. Two UAP56/DDX39B RNA helicases are juxtaposed at each end of the tetramer, which would allow one bivalent ALYREF protein to bridge adjacent helicases and regulate the TREX–mRNA interaction. Our structural and biochemical results suggest a conserved model for TREX complex function that depends on multivalent interactions between proteins and mRNA.

**\*For correspondence:**
clemens.plaschka@imp.ac.at

[†]These authors contributed equally to this work

**Competing interests:** The authors declare that no competing interests exist.

## Introduction

Eukaryotic protein-coding mRNA is matured in the nucleus before it is exported and translated in the cytoplasm. During maturation, mRNA is capped, spliced, and poly-adenylated to form fully processed and packaged messenger ribonucleoprotein complexes (mRNPs) (*Köhler and Hurt, 2007*; *Singh et al., 2015*; *Heath et al., 2016*; *Stewart, 2019*; *Xie and Ren, 2019*). The transcription and export (TREX) complex is recruited during transcription (*Heath et al., 2016*; *Viphakone et al., 2019*) to maturing mRNPs through mRNA and protein interactions at the mRNP 5'-end, splice junctions, and mRNP 3'-end (*Cheng et al., 2006*; *Merz et al., 2007*; *Gromadzka et al., 2016*; *Shi et al., 2017*). TREX subsequently loads the global mRNA-export factor NXF1–NXT1, to license mRNAs for export (*Strässer and Hurt, 2001*; *Köhler and Hurt, 2007*; *Hautbergue et al., 2008*; *Taniguchi and Ohno, 2008*). By chaperoning the nascent mRNA, TREX also inhibits the formation of harmful DNA-RNA hybrids, called R-loops, and protects genome integrity (*Luna et al., 2019*; *Pérez-Calero et al., 2020*).

The TREX complex is found in all eukaryotes and contains the multi-subunit THO complex, the DEXD-box RNA helicase UAP56/DDX39B (yeast Sub2), and an RNA export adapter such as ALYREF (yeast Yra1) (*Strässer et al., 2002*). The human THO complex comprises six subunits, THOC1, −2, −3, −5, −6, and −7, of which four have known counterparts in the yeast *Saccharomyces cerevisiae* (*Sc*): THOC1 (yeast Hpr1), −2 (yeast Tho2), −3 (yeast Tex3), and −7 (yeast Mft1) (*Heath et al., 2016*; *Mitchell et al., 2019*). Additional TREX interactors include SARNP (yeast Tho1), the mammalian protein ZC3H11A and ALYREF-like proteins UIF, LUZP4, POLDIP3, and CHTOP (*Dufu et al., 2010*; *Heath et al., 2016*). In this study we focus on the conserved TREX complex (*Heath et al., 2016*;

**eLife digest** The DNA of human and other eukaryotic cells is stored inside a compartment called the nucleus. DNA carries the genetic code and provides a blueprint for all of the cell's proteins. However, protein production occurs outside the nucleus, in the main body of the cell. To transmit genetic information from one compartment to the other, the DNA sequences are first transcribed into another molecule called messenger RNA, or mRNA for short. Once made, mRNA exits the nucleus and enters the cell's main body to encounter the machinery that translates its sequence into a protein.

Before mRNA can exit the nucleus, it must first undergo a series of modifications, which result in the mRNA molecule being successively bound to specific proteins. Once mRNA has passed through these steps, it is recognized by the transcription-and-export complex, or TREX for short, which is comprised of several proteins. When TREX binds to mRNA, it adds on a final protein which allows the mRNA molecule to be transported out of the nucleus. However, it remained unclear how TREX selects the completed mRNA-protein complexes that are ready for export while at the same time recognizing the wide variety of mRNA molecules produced by cells.

Now, Pühringer and Hohmann et al. have identified the first three-dimensional structure of the core of the human TREX complex using a technique called cryo-electron microscopy. This revealed that the seven proteins of the TREX core assemble into a large complex that has four copies of each protein. The structure suggests that TREX can bind to mRNA and its attached proteins in various ways. These different binding arrangements may help the complex select which mRNA molecules are fully modified and ready to be exported. The structure also sheds light on how mutations in this complex can lead to diseases such as Beaulieu–Boycott–Innes syndrome (BBIS).

This work will help guide future research into the activity of TREX, including how its structure changes when it binds to mRNA and deposits the final transport protein. Identifying these structures will make it easier to design experiments that target specific aspects of TREX activity and provide new insights into how these complexes work.

*Xie and Ren, 2019*): THO–UAP56/DDX39B–ALYREF, and hereafter refer to UAP56/DDX39B as UAP56.

In the current model of mRNA export, the THO complex is recruited to the mRNP and delivers UAP56, which subsequently 'clamps' the mRNA. THO–UAP56 binds the export adapter ALYREF and together they promote loading of the export factor NXF1–NXT1 onto mRNA (*Hautbergue et al., 2008*; *Köhler and Hurt, 2007*; *Strässer and Hurt, 2001*; *Taniguchi and Ohno, 2008*). However, the structural basis for TREX activities remains poorly understood despite active study (*Peña et al., 2012*; *Pérez-Alvarado et al., 2003*; *Ren et al., 2017*; *Shi et al., 2004*; *Xie and Ren, 2019*).

Here, we present the cryo-electron microscopy (cryo-EM) structure of the human THO–UAP56 complex at 3.3 Å resolution. We resolved the THO architecture, its binding mode to UAP56, and can explain mutations linked to human disease (*Heath et al., 2016*). Our results show that THO–UAP56 adopts a tetrameric 28-subunit architecture, suggesting how TREX may bind multiple mRNA and mRNP regions at the same time. Taken together, the data reveal a conserved mechanism for TREX function that depends on multivalent protein–mRNA interactions.

## Results and discussion

To gain structural insights into TREX function, we prepared the human THO complex by co-expressing its six subunits in insect cells (THOC1, −2 residues 1–1203, −3, −5, −6,−7) and added recombinant UAP56 to reconstitute the THO–UAP56 complex (*Figure 1—figure supplement 1a*, Materials and methods). The complex was imaged under cryogenic conditions using a K3 direct electron detector, yielding ~1.6 million single-particle images. Unsupervised 3D particle sorting and focused refinements resulted in cryo-EM densities of THO–UAP56 at nominal resolutions between 3.3 and 4.7 Å (maps A-E) (*Figure 1—figure supplements 1–4*, Materials and methods). The densities enabled us to build a near-complete atomic model of a 14-subunit THO–UAP56 dimer, which assembles into a 28-subunit tetramer with a total molecular weight of 1.8 MDa (*Figures 1a–c* and *2*, *Figure 1—figure supplements 5* and *6*, *Video 1*). The cryo-EM data showed that two THO–

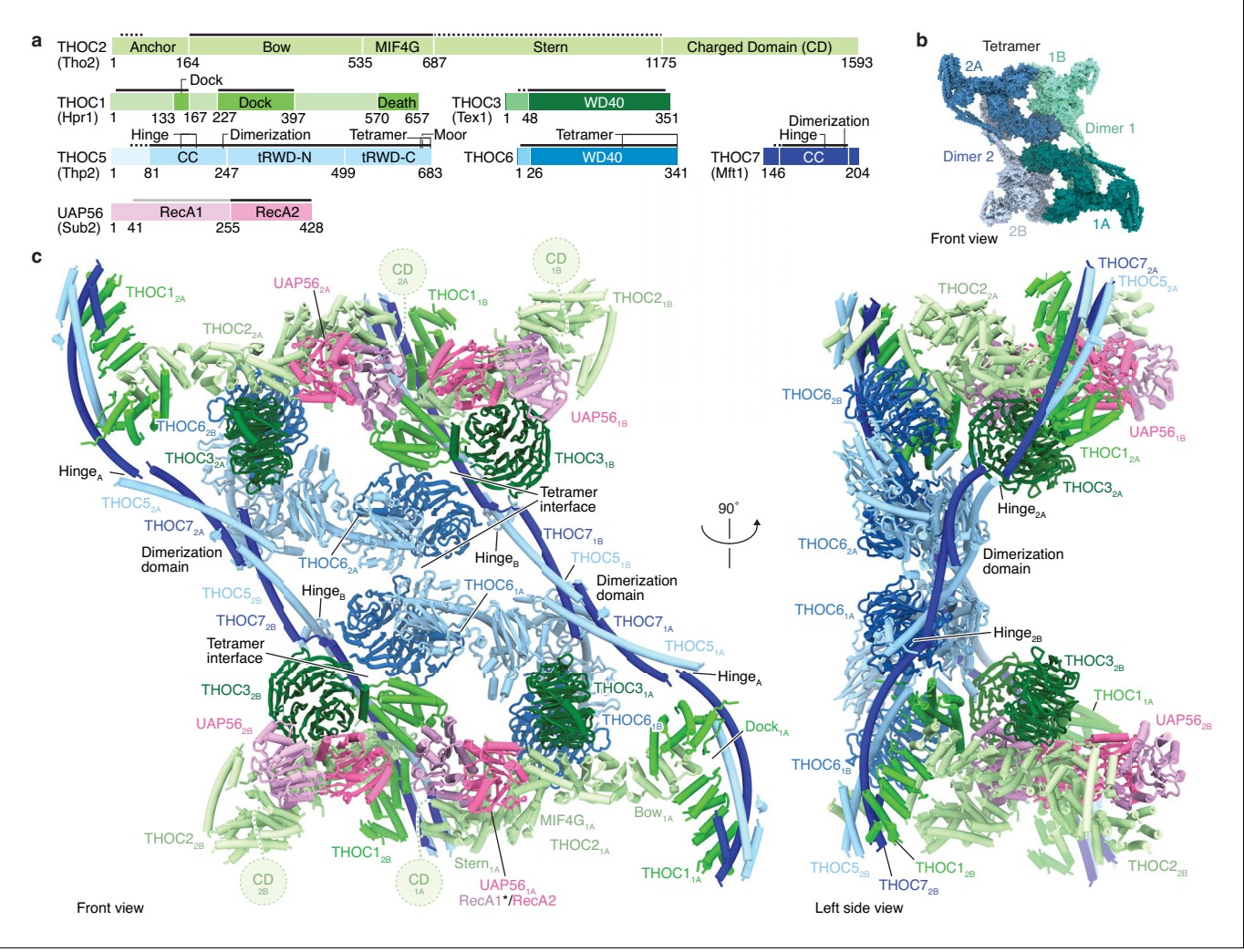

**Figure 1.** Human THO–UAP56 complex structure. (a) Domain organization of human THO–UAP56 complex subunits. Solid and dashed lines indicate atomic and backbone regions of the THO–UAP56 structure, respectively. The UAP56 RecA1 lobe is not observed in the structure and indicated with a light gray line. Orthologous yeast protein names are in parenthesis. Color code used throughout. CC, coiled coil; tRWD-N and -C, N- and C-terminal tandem RWD. (b) Surface representation of the THO–UAP56 tetramer structure. Dimer 1 monomers 1A (dark turquoise) and 1B (light turquoise), and dimer 2 monomers 2A (dark blue), 2B (light blue) are indicated. (c) Human THO–UAP56 structure front and left side views. The UAP56 RecA1 lobe was not visible in the cryo-EM density, but is modeled here by superimposing a UAP56 (yeast Sub2) homology model based on a low-resolution THO–Sub2 crystal structure (*Ren et al., 2017*) (PDB ID 5SUQ) and is indicated with an asterisk. The THO2 CD domain is disordered (*Peña et al., 2012*), was truncated for recombinant THO–UAP56 production, and is shown as a dashed circle.

The online version of this article includes the following figure supplement(s) for figure 1:

**Figure supplement 1.** Biochemical characterization and EM of recombinant and endogenous THO–UAP56 complexes.

**Figure supplement 2.** THO–UAP56 cryo-EM image classification.

**Figure supplement 3.** THO–UAP56 cryo-EM reconstructions.

**Figure supplement 4.** THO–UAP56 cryo-EM data collection and refinement statistics.

**Figure supplement 5.** Gallery of THO–UAP56 subunits and their cryo-EM densities.

**Figure supplement 6.** Back view and surface charge distribution of THO–UAP56.

UAP56 tetramers associate loosely to form an octamer, however, we could not observe direct molecular contacts between the two tetramers (*Figure 1—figure supplement 1c,d*). To determine the in vivo oligomeric state of THO–UAP56, we in addition purified the endogenous complex from human K562 cells and calculated negative stain 2D class averages (*Figure 1—figure supplement 1f–h*). The endogenous THO–UAP56 complex formed a tetramer, indistinguishable from the modeled tetramer structure, suggesting that this state exists in vivo (*Figure 1—figure supplement 1h*).

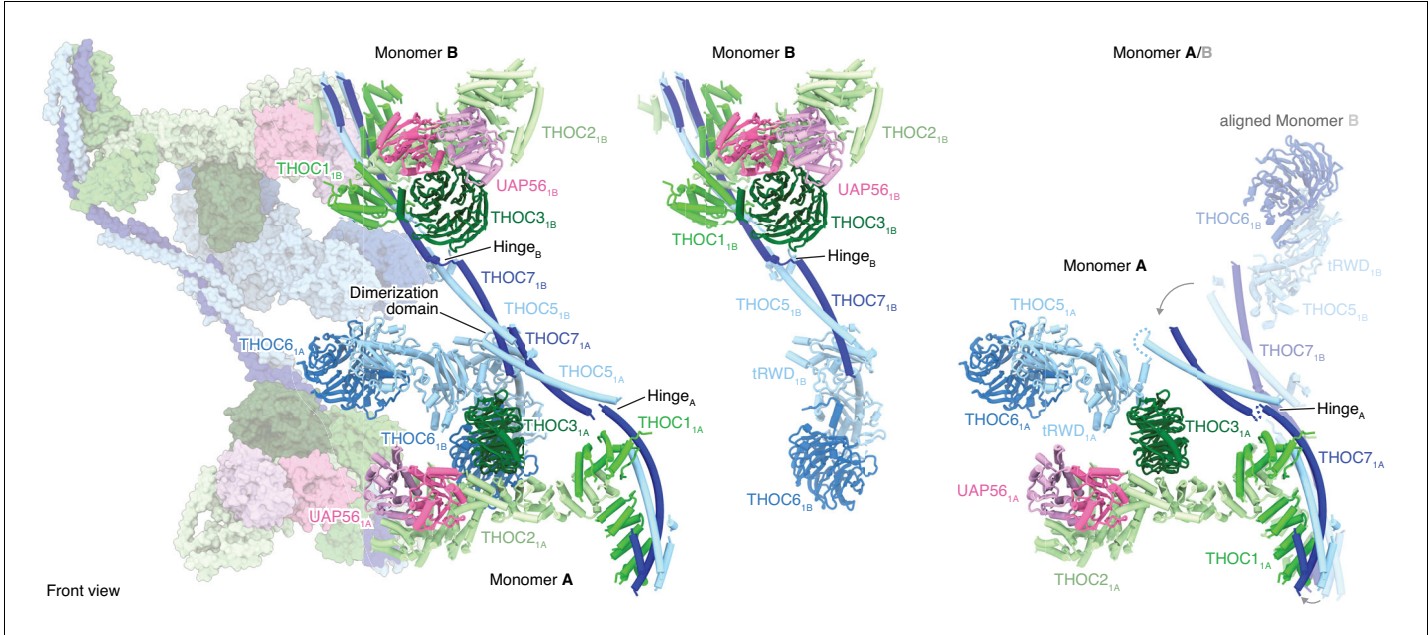

**Figure 2.** THO–UAP56 tetramer, dimer, and monomer organization. Overview of the THO–UAP56 architecture from the front, showing the asymmetric dimers 1 and 2 as ribbons and surfaces, respectively on the left. THO–UAP56 monomers 1B (middle) and 1A (right) are shown as ribbons. THOC5, −6, and −7 assume different positions in monomers A and B, shown in the right panel by superposing monomer 1B (transparent ribbons) on monomer 1A via their THOC2 subunits. This movement is indicated with gray arrows. Colors as in *Figure 1*.

The online version of this article includes the following figure supplement(s) for figure 2:

**Figure supplement 1.** Biochemical probing of THO complex oligomerization, the THO–UAP56 interface, and mapping of disease mutations.

## THO–UAP56 structure

The THO–UAP56 28-subunit tetramer consists of two asymmetric dimers, 1 and 2, each comprising two seven-subunit THO–UAP56 monomers, A and B. Using this nomenclature, the tetramer contains monomers 1A, 1B, 2A, and 2B (*Figures 1b,c* and *2*). Each monomer is partitioned into two regions, one formed by THOC1, −2, –3, and UAP56 and the second by THOC5, −6, and −7. THOC1, −2, –3, and UAP56 adopt the same architecture in all monomers, whereas THOC5, −6, –7 assume different conformations to assemble the dimer via THOC5 and THOC7 and the tetramer via THOC5 and THOC6. THOC5 and THOC7 form a parallel coiled coil, whose C-terminal ends from monomer A and B meet in a four-helix bundle, yielding the dimerization region (*Figures 2* and *3a,b*). The dimer is further stabilized by contacts between the N-terminal THOC5 tandem RWD (tRWD-N) domains from monomers A and B that pack against each other. The THOC5–THOC7 coiled coil is interrupted by a 'hinge' in each monomer, which generates three mobile sub-regions in the 14-subunit THO–UAP56 dimer: monomer A, monomer B, and the above mentioned dimerization region (*Figures 2* and *3a*). Notably, the THOC5–THOC7 coiled coil measures ~ 350 Å across the human dimer, suggesting that distal positioning of monomers A and B may play a role in TREX function.

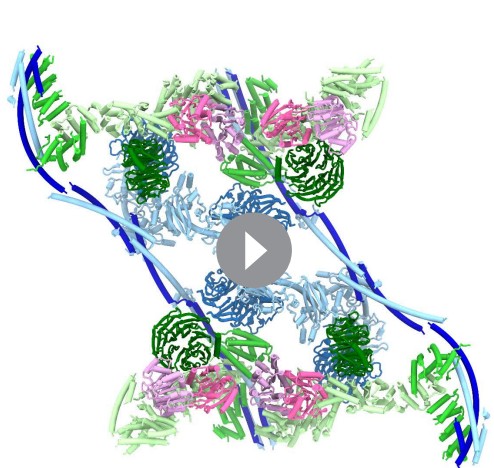

**Video 1.** 360 degree rotation of the human THO–UAP56 complex, colored as in *Figure 1c*.
https://elifesciences.org/articles/61503#video1

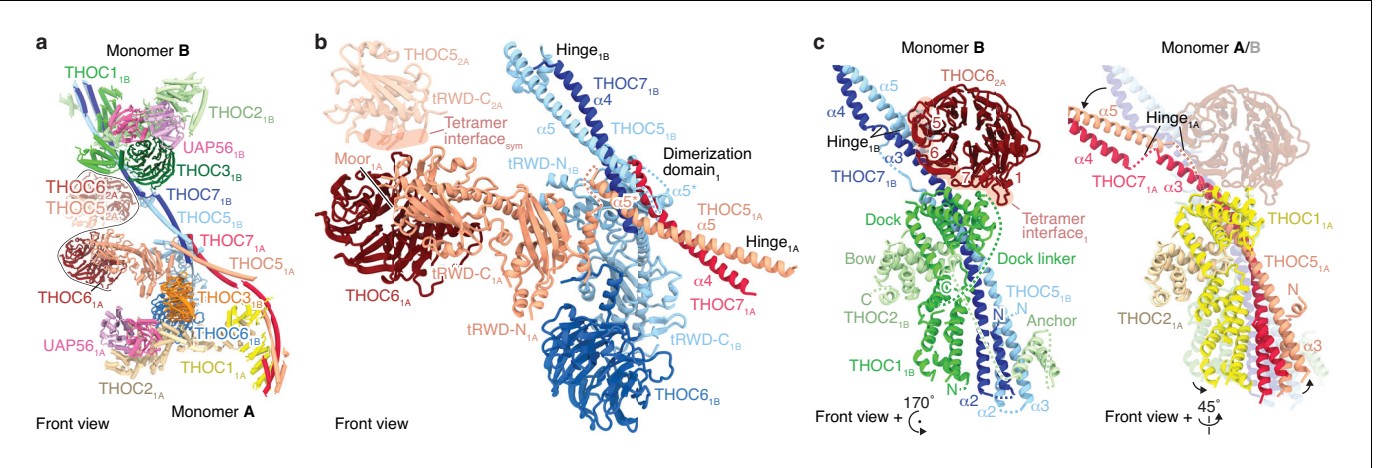

**Figure 3.** Details of THO–UAP56 tetramer and dimer interfaces. (a) Ribbon model of the THO–UAP56 complex dimer 1, indicating dimer and tetramer interfaces. Colors for monomer A subunit UAP56 and monomer B are as in *Figure 1*, monomer A THOC1 (yellow), THOC2 (pale yellow), THOC3 (orange), THOC5 (salmon), THOC6 (maroon), and THOC7 (red). The black line marks the tetramer interface. The THOC5 tRWD-C domain and THOC6 from monomer 2A are shown as transparent ribbons and stabilize the tetramer. (b) THO–UAP56 dimerization domain. Monomer A THOC5 helix α5 and THOC7 α4 make a parallel coiled coil that forms a four-helix bundle with equivalent helices from monomer B. The four-helix bundle is asymmetric (note the α5* position between THOC5$_{1A}$ and THOC5$_{1B}$) due to dimerization of the adjacent THOC5 tRWD-N domains. View and colors as in panel a. (c) The THOC5–THOC7 coiled coil binds THOC1 and the THOC2 anchor helices and adopts different positions in presence (left, monomer 1B) or absence (right, monomer 1A) of THOC6$_{2A}$ from the neighboring THO–UAP56 monomer 2A. Interacting THOC6$_{2A}$ β-propeller blades 1, 5, 6, and seven are labelled. Monomers A and B are superimposed (right) on their THOC2 subunits, and monomer B is transparent. Colors as in panel **a**.

The THO–UAP56 tetramer assembles from dimer 1 and 2 via two interfaces that involve the THOC5 and THOC6 subunits. The first interface is formed by homotypic interactions between the THOC5 tRWD-C domains from monomers 1A and 2A. The second is formed by the THOC6 β-propeller from monomers 1A and 2A that interact with the THOC1, −5, and −7 subunits from monomers 2B and 1B, respectively (*Figure 3a–c*). THOC6 thereby associates with the neighboring dimer and causes THOC1 and the THOC5–THOC7 coiled coil to adopt different positions (*Figure 3c*). This interaction stabilizes the tetramer and at the same time leads to the asymmetry within the THO–UAP56 dimer (*Figure 3c*). In agreement with this architecture, the recombinant THO complex lacking THOC6 assembles into a dimer rather than a tetramer, shown by its slowed migration in sucrose density gradients (*Figure 2—figure supplement 1a*). Deletion of the THOC5 and THOC7 subunits leads to loss of THOC6, as expected, and results in a monomeric THOC1/2/3 complex (*Figure 2—figure supplement 1a*). Nine residues in THOC6 are known to be mutated in human disease (*Heath et al., 2016*), all of which map to the THOC6 β-propeller. These mutations are predicted to destabilize the β-propeller fold or the THOC5–THOC7 interaction and would thereby disrupt THO tetramerization (*Figure 2—figure supplement 1b*). Indeed, four mutations were previously shown to reduce nuclear THOC6 levels and perturb THOC6 interaction with the THO complex (*Mattioli et al., 2019*).

The core of the THO–UAP56 monomer is formed by the THOC2 subunit, which comprises an extended helical repeat with five distinct domains that we name 'anchor', 'bow', 'MIF4G', 'stern', and the disordered 'charged domain' (CD, truncated for recombinant expression) (*Peña et al., 2012*; *Figures 1a* and *4a*). The N-terminal THOC2 anchor helices bind α-helices THOC5 α3 and THOC7 α2 and connect via a disordered linker to the THOC2 bow domain. The THOC2 bow is sandwiched by the helical THOC1 'dock' domain (*Figure 3b*). Hence, the THOC1, −5, and −7 subunits jointly bind THOC2, and THOC1 orients the extended and mobile THOC2 helical repeat (bow, MIF4G, stern) (*Figures 1c* and *2*, *Figure 1—figure supplements 1d* and *3c*). The THOC2 bow and adjacent MIF4G domain bind the THOC3 β-propeller (*Figure 4a*), and the THOC2 MIF4G domain makes additional, extensive contacts to the UAP56 helicase (*Figures 1a* and *4a*).

UAP56 contains two ATPase lobes, RecA1 and RecA2. We observed density for the UAP56 RecA2 lobe, but not RecA1, which remains mobile in the absence of RNA and ATP (*Shi et al., 2004*). Consistent with the structure, THO oligomeric state did not impact UAP56 binding in a pulldown

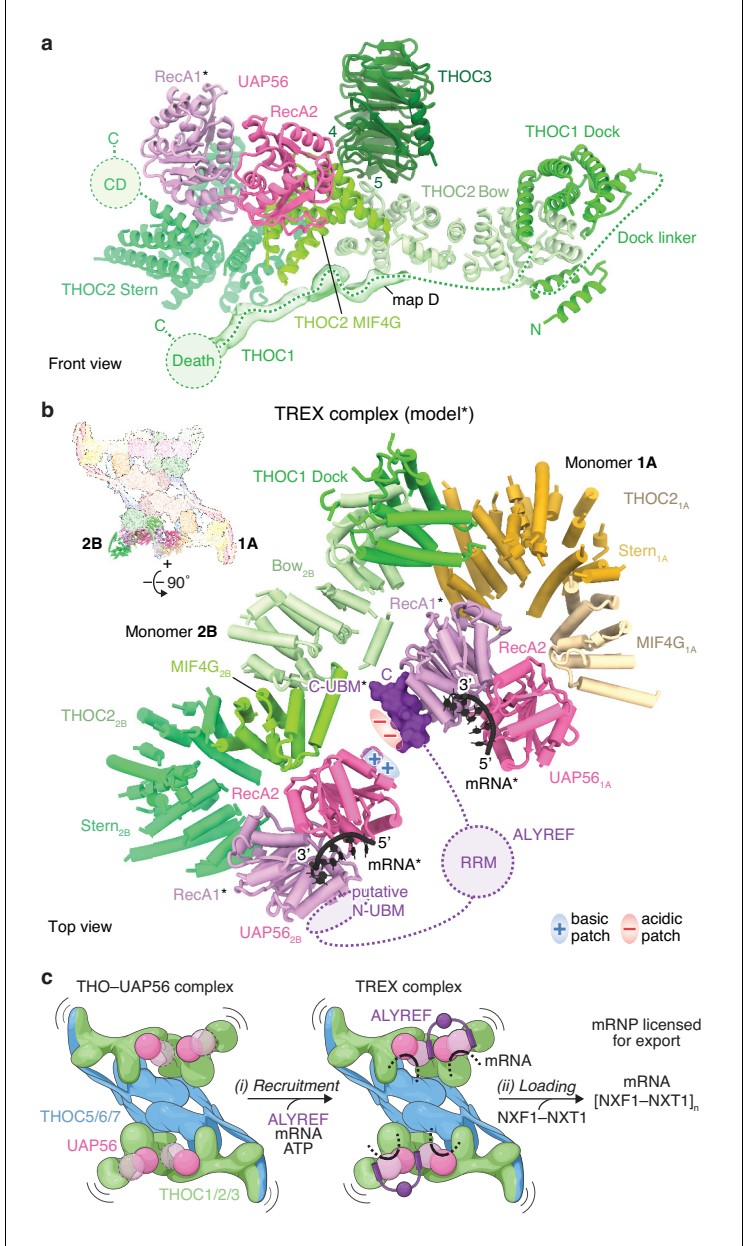

**Figure 4.** THOC2–UAP56 interface and TREX model. (a) Details of THOC2 regions (bow, MIF4G, stern, and CD) and their interactions with THOC1, THOC3, and UAP56. THOC3 β-propeller blades 4 and 5 bind the THOC2 bow domain. The UAP56 N-terminal RecA1 lobe is mobile and modeled as for *Figure 1*, indicated with an asterisk. Unmodeled density (map D) was putatively assigned to THOC1 and binds along the THOC2 bow, MIF4G and stern domains. THOC2 is colored in shades of green, other subunits as in *Figure 1*. (b) TREX–mRNA complex model. The model suggests that ALYREF may license mRNA export by bridging two adjacent UAP56 helicases through (i) its N- and C-UBMs and (ii) charge complementary between acidic and basic residues of one ALYREF UBM, bound to $UAP56_{1A}$, and the juxtaposed $UAP56_{2B}$. The model was obtained by superimposing the human homology model of the yeast Sub2–Yra1–RNA crystal structure (*Ren et al., 2017*) (PDB ID 5SUP) on the UAP56 RecA2 lobe, in each monomer. The asterisk indicates modeled elements (UAP56 RecA1, mRNA, ALYREF). THO–UAP56 monomers 1A and 2B are positioned in proximity, according to 3D variability analysis (*Figure 4—figure supplement 1d,e*). THOC3, −5, −7, the $THOC2_{1A}$ bow, $THOC1_{1A}$, and the $THOC1_{2B}$ N-terminus were omitted for clarity. Colors as in *Figure 3*, ALYREF (violet), RNA (black). RRM, RNA-recognition motif; N- and C-UBM, N- and C-terminal UAP56-binding motifs. (c) Cartoon schematic of TREX-dependent mRNA-export licensing. Multivalent interactions between THO–UAP56, mRNA, and mRNP proteins may be required for (i) mRNP recruitment and mRNA binding. ALYREF may bridge two proximal UAP56 helicases in the assembled TREX–mRNA complex, to

*Figure 4 continued on next page*

*Figure 4 continued*

facilitate (ii) loading of multiple NXF1/NXT1 export factor complexes onto nascent mRNA and license it for export. The curved lines indicate flexibility. The UAP56 RecA1 lobe is mobile in absence of mRNA, indicated by dashed lines and the transparent surface.

The online version of this article includes the following figure supplement(s) for figure 4:

**Figure supplement 1.** Details of THO–UAP56 flexibility and the UAP56–ALYREF interaction.

**Figure supplement 2.** ATPase assay and structural comparisons of the DEXD-box RNA helicase–MIF4G domain interaction.

**Figure supplement 3.** Revised yeast THO–Sub2 model and conservation of the THO architecture.

---

experiment (*Figure 2—figure supplement 1c*), whereas point mutations in the THOC2 MIF4G domain at the UAP56 interface did impair UAP56 binding (*Figure 2—figure supplement 1d,e*). The THOC2 MIF4G domain adjoins the curved THOC2 stern domain, which in turn connects to the disordered CD implicated in nucleic acid binding (*Peña et al., 2012*). Near the THOC2 stern, we observed weak density for the THOC1 C-terminus (*Figure 4a*), which can bind the mRNA-export factor in yeast and humans (*Hobeika et al., 2007*; *Viphakone et al., 2012*). The close positioning of UAP56, the THOC1 C-terminus, and the THOC2 CD suggests that this region is involved in export factor loading. Supporting this model, we observed that the recombinant core THOC1/2/3–UAP56 complex is sufficient to bind the NXF1–NXT1 export factor in vitro (*Figure 2—figure supplement 1f*). Previous data showed that the isolated THOC5 subunit can also bind NXF1–NXT1 in vitro (*Katahira et al., 2009*; *Viphakone et al., 2012*). However, THOC5 did not contribute to NXF1–NXT1 binding in the context of multi-subunit THO–UAP56 complexes (*Figure 2—figure supplement 1f*), consistent with its distant location from the putative export factor loading site in the THO–UAP56 structure (*Figures 1c* and *4a*).

The THO–UAP56 tetramer is ~260 Å long, ~290 Å high, and ~150 Å wide. Subunits involved in mRNA and export factor binding are located at the ends of the complex (THOC1, −2, –3, and UAP56), away from those involved in oligomerization (THOC5, −6, and −7) (*Figures 1c* and *2*, *Figure 1—figure supplement 6a*). Whereas all THO subunits are essential genes (*Dempster et al., 2019*), the requirement for THOC5, −6, and −7 suggests that THO oligomerization may be needed for normal function. The extended shape and flexibility of the THO–UAP56 tetramer (*Figure 4—figure supplement 1d*) could allow for multiple interactions with spatially distant mRNA regions and mRNP maturation marks, such as the cap-binding complex or exon-junction complex (*Cheng et al., 2006*; *Merz et al., 2007*) and facilitate NXF1–NXT1 loading from multiple sites.

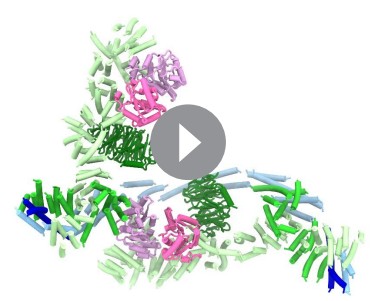

**Video 2.** 360 degree rotation of the yeast THO–Sub2 complex, colored as in *Figure 4—figure supplement 3d*. https://elifesciences.org/articles/61503#video2

## TREX complex and ALYREF function

ALYREF (yeast Yra1) is part of the conserved TREX complex (*Chávez et al., 2000*; *Strässer et al., 2002*), is required for transfer of mRNA to the export factor NXF1–NXT1 (*Hautbergue et al., 2008*; *Taniguchi and Ohno, 2008*), and is essential for viability in yeast (*Strässer and Hurt, 2000*; *Stutz et al., 2000*) and humans (*Dempster et al., 2019*). ALYREF contains two UAP56-binding motifs (UBMs) at its N- and C-terminus that can separately bind UAP56 (*Hautbergue et al., 2009*; *Figure 4—figure supplement 1a,b*). Between the UBMs is an RG-rich region, followed by a central RNA-recognition motif (RRM) domain, and a second RG-rich region (*Gromadzka et al., 2016*). To gain insight into ALYREF function, we prepared a homology model of UAP56 bound to RNA, an ATP-analog, and the C-terminal ALYREF UBM (C-UBM) based

on a yeast crystal structure (*Ren et al., 2017*). We then superimposed this model onto the human THO–UAP56 tetramer structure via the four UAP56 RecA2 lobes (*Figure 4b*, *Figure 4—figure supplement 1d,e*). The resultant model reveals how ALYREF may function in the TREX complex bound to mRNA.

First, the model suggests that a single ALYREF protein may reach across the two THO dimers to bind two juxtaposed UAP56 helicases. The ALYREF N- and C-UBMs would each bind one UAP56 RecA1 lobe (*Hautbergue et al., 2009*), located 20 Å apart, from THO–UAP56 monomers 1A and 2B (or 2A and 1B) (*Figure 4b*). Alternatively, ALYREF could bridge two UAP56 proteins within one dimer, which are ~200 Å apart, or possibly multiple THO–UAP56 tetramers. Bridging by ALYREF can explain why deletion of either N- or C-UBM leads to growth defects in yeast (*Strässer and Hurt, 2001*; *Zenklusen et al., 2001*) and suggests that the UBMs and associated RG-rich domains contribute to the avidity of TREX–mRNA binding, consistent with biochemical data (*Hautbergue et al., 2008*; *Ren et al., 2017*). Furthermore, RNA binding by UAP56 (monomers 1A and 2A) would orient the mobile RecA1 lobe and project conserved negatively charged residues of the interacting ALYREF UBM towards a conserved patch of positively charged residues of the juxtaposed UAP56 RecA2 lobe (from monomers 1B and 2B) (*Figure 4b*). This RecA2 surface is frequently bound by regulators in other DEXD-box helicases (*Linder and Jankowsky, 2011*; *Buchwald et al., 2013*; *Sharif et al., 2013*; *Mathys et al., 2014*; *Figure 4—figure supplement 1g*) and suggests how a single UBM could be sufficient for partial ALYREF function (*Strässer and Hurt, 2001*; *Zenklusen et al., 2001*; *Golovanov et al., 2006*; *Gromadzka et al., 2016*). Notably, these two models of ALYREF function are not mutually exclusive and collectively suggest that the ALYREF UBMs bridge UAP56 helicases, which may correctly position other ALYREF regions for export factor loading.

The functionally related ALYREF-like proteins contain one (LUZP4, POLDIP3, UIF) or two (CHTOP) closely spaced UBMs (*Gromadzka et al., 2016*; *Hautbergue et al., 2009*; *Viphakone et al., 2015*; *Figure 4—figure supplement 1c*), and thus are unlikely to bridge two helicase RecA1 lobes in the THO–UAP56 structure (*Figure 4—figure supplement 1c*). Nevertheless, we speculate that the THO–UAP56 tetramer increases the affinity of mRNA-export adaptors to their target mRNAs through avidity, by providing multiple UAP56 binding sites in close proximity.

UAP56 is a member of the DEXD-box RNA helicase family. These helicases function as 'clampases' that couple ATP hydrolysis to RNA release and can be regulated by interacting proteins (*Linder and Jankowsky, 2011*). Biochemical data from yeast (*Ren et al., 2017*) and humans (*Taniguchi and Ohno, 2008*; *Shen et al., 2007*) show that ALYREF weakly promotes UAP56 ATPase activity, and the yeast THO complex further stimulates this (*Ren et al., 2017*). We observed that the human THOC1/2/3 core complex is sufficient to promote UAP56 ATPase activity in vitro, consistent with our structure and other MIF4G–DEXD-box helicase systems (*Hilbert et al., 2011*; *Montpetit et al., 2011*; *Figure 4—figure supplement 2a,b*). Taken together, we propose that THO and ALYREF regulate the activity of two bridged UAP56 helicases within the TREX–mRNA complex to control export factor loading onto mRNA (*Hautbergue et al., 2008*; *Taniguchi and Ohno, 2008*).

## Conserved THO–UAP56 dimerization

To gain insights into conservation of the TREX complex, we re-analyzed a previously reported 6.0 Å resolution crystal structure of the six-subunit yeast *Sc* THO–Sub2 complex (*Ren et al., 2017*; *Figure 4—figure supplement 3*, Materials and methods). The yeast structure was proposed to be a six-subunit monomer, within which a single Tex1 (THOC3) and a single Sub2 (UAP56) subunit were assigned, but not Tho2, Hpr1, Thp2, or Mft1. Surprisingly, our revised yeast model revealed that the crystal contained a 12-subunit THO–Sub2 dimer, which could better explain the published electron density and phosphotungsten cluster positions (*Figure 4—figure supplement 3a–c*, *Video 2*). We could assign two copies of Tho2, Hpr1, the Thp2–Mft1 coiled coil, and place one additional copy of the Tex1 and Sub2 subunits based on (i) comparisons to the human THO–UAP56 structure, (ii) the known homologies of yeast Hpr1, Tho2, Tex1, and Mft1 with human THOC1, −2,–3, and −7, and (iii) the predicted Thp2 coiled coil (*Söding et al., 2005*; *Figure 4—figure supplement 3d*, Materials and methods). The revised model allowed us to identify Thp2 and confirm Mft1 as the respective yeast homologs of the THOC5 and THOC7 coiled coils. This indicates an unexpectedly high degree of structural conservation of the five-subunit THO complex monomer (THOC1, −2, –3, −5, –7) and its dimerization via coiled coils. Notably, the THO–Sub2 model shows that the two Sub2 helicases are located ~80 Å apart. Both bind to their respective Tho2 MIF4G domain in monomers A and B,

analogous to the human THOC2–UAP56 interaction. Thus, also in yeast a single Yra1 could bridge two Sub2 helicases via its N- and C-UBM regions, mirroring human ALYREF (*Figure 4—figure supplement 3d*).

### Model of mRNA-export licensing

Our results suggest a unified model for TREX function through multimerization of the THO–UAP56 complex (*Figure 4c*). THO–UAP56 forms a dimer in yeast (*Figure 4—figure supplement 3*) and a constitutive tetramer in humans, mediated by the additional THOC5 tRWD domain and THOC6 subunit (*Figure 1*, *Figure 1—figure supplement 1f–h*, *Figure 4—figure supplement 3f*). In humans, THO–UAP56 tetramerization may have evolved in response to an increased complexity in gene architecture and mRNP composition (*Singh et al., 2015*). The conserved THO–UAP56 multimerization, extended architecture, and flexibility (*Figure 1—figure supplement 1b–d*, *Figure 4—figure supplement 1d*) indicate that one complex can bind multiple mRNP maturation marks and mRNA regions simultaneously. Subsequent to THO–UAP56 recruitment to the mRNP, a single ALYREF (or yeast Yra1) molecule could bridge two juxtaposed UAP56–mRNA complexes and, potentially, multimerize itself through its RG-rich regions (*Gromadzka et al., 2016*). In the assembled TREX–mRNP complex UAP56 ATPase activity is stimulated and, together with ALYREF or alternative export adaptors, may load the export factor NXF1–NXT1 from multiple sites within the complex. Taken together, the data support a model where requirements for both the selectivity and efficiency of mRNP recognition and mRNA-export licensing are fulfilled through multivalent interactions between proteins and mRNA.

## Materials and methods

**Key resources table**

| Reagent type (species) or resource | Designation | Source or reference | Identifiers | Additional information |
|---|---|---|---|---|
| Gene (*H. sapiens*) | THOC1 | Uniprot | Q96FV9 | |
| Gene (*H. sapiens*) | THOC2 | Uniprot | Q8NI27 | |
| Gene (*H. sapiens*) | THOC3 | Uniprot | Q96J01 | |
| Gene (*H. sapiens*) | THOC5 | Uniprot | Q13769 | |
| Gene (*H. sapiens*) | THOC6 | Uniprot | Q8NI27 | |
| Gene (*H. sapiens*) | THOC7 | Uniprot | Q6I9Y2 | |
| Gene (*H. sapiens*) | UAP56/DDX39B | Uniprot | Q13838 | |
| Gene (*H. sapiens*) | NXF1 | Uniprot | Q9UBU9 | |
| Gene (*H. sapiens*) | NXT1 | Uniprot | Q9UKK6 | |
| Gene (*H. sapiens*) | NUP214 | Uniprot | P35658 | |
| Strain, strain background (*Escherichia coli*) | DH10EmBacY | Geneva Biotech | DH10EmBacY | Strain propagated locally from commercial source stock |

*Continued on next page*

*Continued*

| Reagent type (species) or resource | Designation | Source or reference | Identifiers | Additional information |
|---|---|---|---|---|
| Strain, strain background (*E. coli*) | BL21(DE3)RIL | Agilent | 230245 | Strain propagated locally from commercial source stock |
| Cell line (Spodoptera frugiperda) | Sf9 | Thermo Fisher Scientific | IPLB-Sf-21-AE; RRID:CVCL_0549 | Strain propagated locally from commercial source stock |
| Cell line (Trichoplusia ni) | High Five | Thermo Fisher Scientific | BTI-TN-5B1-4; RRID:CVCL_C190 | Strain propagated locally from commercial source stock |
| Cell line (Lenti-X) | Lenti-X | Takara | Cat. #632180 | |
| Cell line (K562) | K562 | DSMZ | ACC 10, RRID:CVCL_0004 | |
| Antibody | anti-THOC1 (Polyclonal rabbit) | Merk | HPA019096 | (dilution 1:500) |
| Antibody | anti-mouse IgG-HRP (Polyclonal goat) | Promega | W4021 | (dilution 1:10000) |
| Recombinant DNA reagent | pACEBac1 | Geneva Biotech | pACEBac1 | |
| Recombinant DNA reagent | pOPINB | Addgene | #41142 | |
| Recombinant DNA reagent | Lentiviral vector | Addgene | #31485 | |
| Software, algorithm | hhpred | doi:10.1093/nar/gki408 | RRID:SCR_010276 | https://toolkit.tuebingen.mpg.de/tools/hhpred |
| Software, algorithm | WARP | doi: 10.1038/s41592-019-0580-y | | http://www.warpem.com/warp/ |
| Software, algorithm | cryoSPARC | doi:10.1038/nmeth.4169 | RRID:SCR_016501 | https://cryosparc.com/ |
| Software, algorithm | Relion | doi:10.7554/eLife.42166 | RRID:SCR_016274 | https://www3.mrc-lmb.cam.ac.uk/relion/ |
| Software, algorithm | ResMap | doi:10.1038/nmeth.2727 | | http://resmap.sourceforge.net |
| Software, algorithm | Modeller | doi:10.1002/cpbi.3 | RRID:SCR_008395 | https://salilab.org/modeller/ |
| Software, algorithm | COOT | doi:10.1107/S0907444910007493 | RRID:SCR_014222 | https://www2.mrc-lmb.cam.ac.uk/personal/pemsley/coot/ |
| Software, algorithm | Phenix | doi:10.1107/S2059798318006551 | RRID:SCR_014224 | https://www.phenix-online.org/ |
| Software, algorithm | USCF ChimeraX | doi:10.1002/pro.3235 | RRID:SCR_015872 | https://www.rbvi.ucsf.edu/chimerax/ |
| Software, algorithm | Pymol | Schrödinger LLC, 2020 | RRID:SCR_000305 | https://pymol.org/2/ |

## Vectors and sequences

For heterologous co-expression of the human 6-subunit THO complex in insect cells, the open reading frames (ORFs) of THOC1, −2 residues 1–1203, −3, −5, −6, −7 with an N-terminal 10xhistidine tag on THOC2, an N-terminal 3xV5 tag on THOC1, and a TwinStrepII tag on THOC3 were cloned into a modified pACEBac1 vector (Geneva Biotech) by Golden Gate cloning (*Engler et al., 2008*). THOC2 was truncated for co-expression at its disordered C-terminus (residues 1204–1593) for an improved biochemical behavior (*Figure 1—figure supplement 1a*). The human UAP56 ORF was cloned into a pOPINB vector with an N-terminal 6xhistidine tag for expression in *E. coli*. For endogenous purification of THO–UAP56 from human K562 cells, the THOC1 cDNA and a C-terminal 3C-AID-GFP tag were cloned into a lentiviral vector backbone (Addgene plasmid #31485), yielding a plasmid containing pRRL-SFFV-THOC1-3C-AID-GFP. The human heterodimer NXF1–NXT1 ORFs with an N-terminal 10xhistidine-3xflag tag on NXF1 and, separately, the human NUP214 FG-repeat (residues 1916–2033) ORF with an N-terminal Maltose-binding protein fusion and C-terminal 10xhistidine tag were cloned into modified pACEBac1 vectors as done for the THO complex.

## Preparation and reconstitution of the THO-UAP56 complex

Recombinant THO complex was co-expressed in insect cells using a plasmid containing all six subunits (THOC1, −2 residues 1–1203, −3, −5, −6, −7). The THO plasmid was electroporated into DH10EMBacY cells to generate bacmids (*Trowitzsch et al., 2010*) that were then transfected into *Spodoptera frugiperda* Sf9 cells to generate a V0 virus. The V0 virus was further amplified in Sf9 cells to yield V1 virus. For protein expression, 750 mL *Trichoplusia ni* Hi5 cells at a density of $1 \times 10^6$ cells were infected with 1 mL V1 virus and harvested after 3 days by centrifugation. Harvested cells were resuspended in buffer A (50 mM Tris-HCl pH 8.0, 300 mM NaCl, 5% (w/v) glycerol, 20 mM imidazole, 1 mM dithiothreitol (DTT), 0.5 mM PMSF, cOmplete EDTA-free protease inhibitor cocktail (Roche)) and lysed by sonication. The lysate was clarified by ultracentrifugation and the supernatant loaded on a HisTrap HP 5 mL column (GE Healthcare), equilibrated in buffer B (10 mM Tris-HCl pH 8.0, 300 mM NaCl, 5% (w/v) glycerol, 20 mM imidazole, 1 mM DTT). The column was washed with a linear gradient from 20 mM to 100 mM imidazole and eluted with buffer B containing 350 mM imidazole. Peak fractions were diluted 1:1 with buffer B lacking NaCl and imidazole and applied to anion exchange using a HiTrapQ HP 5 mL column (GE Healthcare), equilibrated in buffer C (25 mM HEPES pH 7.9, 150 mM NaCl, 5% (w/v) glycerol, 2.5 mM DTT). The complex was eluted with a linear gradient of buffer C from 150 to 1000 mM NaCl. Fractions containing the THO complex were concentrated, loaded on a HiLoad 16/600 Superdex 200 pg column (GE Healthcare) equilibrated in buffer D (25 mM HEPES pH 7.9, 250 mM NaCl, 5% (w/v) glycerol, 1 mM TCEP). The purified THO complex was concentrated to 4 mg mL$^{-1}$, flash frozen and stored at −80°C. Variant THO complexes, lacking subunits THOC6 or containing mutations, were purified as described above. The THOC1/2/3 complex was purified as above, except that buffer A and B contained 500 mM NaCl, the complex was eluted from the HisTrap HP 5 mL column using a linear gradient from 0–50% with buffer B containing 500 mM NaCl and 250 mM Imidazole, and that buffer D lacked TCEP. The identity of all THO subunits was confirmed by mass spectrometry.

Full-length human UAP56 was expressed in *E. coli* BL21 DE3 RIL cells grown in autoinduction media at 37°C for 16 hr. Cells were lysed by sonication in buffer E (25 mM HEPES pH 7.9, 500 mM NaCl, 5% (w/v) glycerol, 20 mM imidazole, 1 mM DTT, 0.5 mM PMSF, 0.1% (v/v) Tween-20, cOmplete EDTA-free protease inhibitor cocktail). The supernatant was clarified by centrifugation and loaded on a HisTrap HP 5 mL column, equilibrated in buffer F (25 mM HEPES pH 7.9, 500 mM NaCl, 5% (w/v) glycerol, 20 mM imidazole, 1 mM DTT). The column was washed with buffer F and eluted with a linear gradient from 20 to 250 mM imidazole. The 6xhistidine tag was cleaved by PreScission protease at 4°C overnight, and UAP56 was then diluted to 50 mM NaCl using buffer F lacking NaCl and imidazole and applied to anion exchange using a HiTrapQ HP 5 mL, equilibrated in buffer G (25 mM HEPES pH 7.9, 50 mM NaCl, 5% (w/v) glycerol, 2.5 mM DTT). The protein was eluted with a linear gradient from 50 to 400 mM NaCl. Fractions containing UAP56 were concentrated and loaded on a HiLoad 16/600 Superdex 75 pg column, equilibrated in buffer H (25 mM HEPES pH 7.9, 100 mM NaCl, 5% (w/v) glycerol, 2.5 mM DTT). Purified UAP56 was concentrated to 11 mg mL$^{-1}$, flash frozen and stored at −80°C.

To reconstitute the THO–UAP56 complex, the THO complex was mixed with a two-fold molar excess of UAP56 and incubated at 20℃ for 30 min in buffer I (25 mM HEPES pH 7.9, 100 mM NaCl, 5% (w/v) glycerol, 1 mM TCEP). We used GraFix (*Kastner et al., 2008*) to obtain a homogenous and crosslinked THO–UAP56 complex for EM studies. 90 µg of THO–UAP56 complex were loaded onto a 4 mL 10–40% (w/v) sucrose gradient in buffer J (25 mM HEPES pH 7.9, 50 mM KCl, 1 mM TCEP, 0–0.05% glutaraldehyde) and ultracentrifuged at 91,100 x g in a SW60 Ti swing bucket rotor (Beckman) for 20 hr 30 min at 4℃. Peak fractions were pooled and the crosslinking reaction was quenched for 15 min using a final concentration of 50 mM Lysine. The sample was then buffer-exchanged using Zeba Spin 7K MWCO Desalting Columns that were equilibrated in buffer K (25 mM HEPES pH 7.9, 50 mM KCl, 2% (w/v) glycerol, 1 mM TCEP). The sample was concentrated using a Vivaspin 500 centrifugal concentrator (MWCO 100 kDa) and immediately used for EM studies.

NXF1–NXT1 virus generation and co-expression in insect cells were performed as for the THO complex (see above). Cells were harvested by centrifugation and resuspended in buffer L (50 mM HEPES pH 7.9, 300 mM NaCl, 10% (w/v) glycerol, 1 mM DTT, cOmplete EDTA-free protease inhibitor cocktail) before sonication. The cell lysate was ultracentrifuged and the supernatant applied to a HisTrap HP 5 mL column, equilibrated in buffer M (25 mM HEPES pH 7.9, 300 mM NaCl, 10% (w/v) glycerol, 30 mM imidazole, 1 mM DTT). The column was washed with 45 mM imidazole and eluted with buffer M containing 300 mM imidazole. Peak fractions were diluted 1:2 with buffer M lacking imidazole and containing 5% (w/v) glycerol and 10 mM NaCl, and were loaded on a HiTrap Heparin HP 5 mL column (GE Healthcare), equilibrated in buffer N (25 mM HEPES pH 7.9, 50 mM NaCl, 5% (w/v) glycerol, 1 mM DTT). The complex was eluted with a linear gradient of buffer N from 50 to 1000 mM NaCl. Fractions containing the NXF1–NXT1 heterodimer were concentrated to 1 mg mL$^{-1}$, flash frozen, and stored at −80℃.

The NUP214 FG-repeat (residues 1916–2033) was expressed in insect cells as done for the THO complex above. The harvested cells were lysed by sonication in buffer A, and applied to a HisTrap HP 5 mL column, equilibrated in buffer B. The column was washed with buffer B and then with buffer B containing 600 mM NaCl. The protein was eluted in buffer B containing 300 mM imidazole and dialyzed overnight against 1 L buffer C containing 200 mM NaCl. The dialyzed protein was loaded onto a HiLoad 16/600 Superdex 200 pg column, equilibrated in buffer C containing 200 mM NaCl. Fractions containing the NUP214 FG-repeat protein were pooled, concentrated to 14 mg mL$^{-1}$, flash frozen, and stored at −80℃.

## Endogenous THOC1 overexpression and nuclear extract preparation

For endogenous purification of THO–UAP56, lentiviral particles carrying the THOC1-3C-AID-GFP construct were generated using Lenti-X cells (Takara) by polyethylenimine transfection (Polysciences) of the viral plasmid and helper plasmids pCMVR8.74 (Addgene plasmid #22036) and pCMV-VSV-G (Addgene plasmid #8454), according to standard procedures. K562 (DSMZ) cells were infected at limiting dilutions and GFP-positive cells were isolated using a BD FACSAria III cell sorter (BD Biosciences). Viral integration was confirmed by immunoblotting for THOC1 and GFP (*Figure 1—figure supplement 1f*). To prepare nuclear extract (NE), 15 L of human K562 cells were grown to a density of $1 \times 10^6$ cells mL$^{-1}$ at 37℃, stirred at 80 rpm and 5% $CO_2$. The NE was prepared as previously described (*Mayeda and Krainer, 1999*) and dialyzed against buffer O (20 mM HEPES, pH 7.9, 100 mM KCl, 20% (w/v) glycerol, 0.2 mM EDTA, 2 mM DTT). Lenti-X (Takara) and K562 (DSMZ) cells tested negative for *mycoplasma*.

## Western blot

NE from wild type and THOC1-3C-AID-GFP containing human K562 cells was diluted 1:1 with buffer P (40 mM HEPES pH 7.9, 15 mM KCl, 6 mM MgCl$_2$, 2 mM DTT). For immunoprecipitation, the NE was first incubated for 1 hr at 4℃ with or without 2 µg Benzonase per mL NE, as indicated in *Figure 1—figure supplement 1f*, and then incubated on a rotating wheel for 2.5 hr at 4℃ with GFP-Trap Agarose resin (Chromotek) that was previously equilibrated with buffer Q (20 mM HEPES pH 7.9, 100 mM KCl, 2 mM MgCl$_2$, 8% (w/v) glycerol, 0.05% (v/v) Igepal CA-630, 1 mM DTT). After five washes, the beads were eluted by boiling and the samples were applied to SDS–PAGE, transferred onto a PVDF membrane (ThermoScientific) and probed with anti-THOC1 primary antibody (HPA019096, Merk, dilution 1:500). Goat anti-mouse IgG-HRP (W4021, Promega, dilution 1:10000)

was used as a secondary antibody. Antibody detection was performed with Amersham ECL Select Western Blotting Detection Reagent (GE Healthcare) and a ChemiDoc MP imaging system (Bio-Rad Laboratories).

## Endogenous THO–UAP56 purification

The THOC1-3C-AID-GFP K562 NE was treated with 1 µg Benzonase per mL NE for 12 hr at 4°C, to release THO–UAP56 complexes from the pellet fraction and substantially increase the total yield. THOC1 was immunoprecipitated as described above (Western blot) and eluted for 2 hr at 4°C with 3C PreScission Protease. The eluate was loaded onto a 10–50% w/v sucrose step gradient containing 0–0.05% glutaraldehyde in buffer T (20 mM HEPES pH 7.9, 100 mM KCl, 2 mM MgCl$_2$, 2 mM TCEP), and centrifuged for 16 hr at 105,600 x g in a SW60 Ti rotor (Beckman coulter). Fractions containing the THO–UAP56 complex were quenched for 15 min using a final concentration of 50 mM lysine, pooled, concentrated in a 0.5 mL 100 kDa MWCO Amicon concentrator (Sigma) and immediately used for EM grid preparation.

## ATPase assay

We measured steady-state UAP56 ATPase activity using a NADH-coupled ATPase assay, essentially as described (*Montpetit et al., 2012*). The assay was set up at the final concentrations of 5 U/mL rabbit muscle pyruvate kinase, Type III (Sigma-Aldrich), 5 U/mL rabbit muscle L-lactic dehydrogenase, Type XI (Sigma-Aldrich), 500 µM phosphoenolpyruvate and 50 µM NADH. Reactions were prepared in buffer R (25 mM HEPES pH 7.9, 100 mM NaCl, 25 mM KCl, 10 mM MgCl$_2$, 5% (w/v) glycerol, 1 mM ATP) and contained the indicated combinations of proteins and RNA at the following concentrations: 2 µM UAP56, 2 µM THOC1/2/3, 10 µM poly-uridine 15 nucleotide RNA. The decay of NADH emission signal was monitored over time at 37°C in a PHERAstar FS (BMG LABTECH), using a 0.03–100 µM NADH dilution series as calibration standard. Average UAP56 ATPase rates from triplicate experiments were calculated from linear slopes of NADH decay as hydrolyzed molecules of ATP s$^{-1}$ per enzyme.

## Pulldown assay

To test whether THO complex oligomeric state influences the THO–UAP56 interaction, we immobilized equimolar amounts of the recombinant THOC1/2/3 complex (3.3 µg), THO$^{\Delta THOC6}$ complex (4.5 µg) and THO complex (5 µg) via the 10xhistidine tag on THOC2 on magnetic nickel beads (Promega) together with 1 µg recombinant human UAP56 (two-fold molar excess over THO) for one hour at 4°C in buffer S (25 mM HEPES pH 7.9, 50 mM KCl, 5% (w/v) glycerol, 20 mM imidazole, 0.01% (v/v) Igepal CA-630). The beads were washed five times with 1 mL buffer S, and proteins were eluted for one hour at 20°C using 500 mM imidazole in buffer S and visualized by SDS-PAGE (Coomassie blue) (*Figure 2—figure supplement 1c*). To probe the THOC2–UAP56 interface, 5 µg of THO complex or three variants (THO$^{M1}$: THOC2 Y551A, K554S, R555S, K558S; THO$^{M2}$: THOC2 K589A, Y590S, N592A; THO$^{M3}$: combined THO$^{M1}$ and THO$^{M2}$ mutants) were immobilized, incubated with 1 µg UAP56, washed in buffer T (25 mM HEPES pH 7.9, 100 mM KCl, 5% (w/v) glycerol, 20 mM imidazole, 0.01% (v/v) Igepal CA-630), eluted and visualized by SDS-PAGE (Coomassie blue) (*Figure 2—figure supplement 1e*). To assess binding between the NXF1–NXT1 export factor and the THO complex, we immobilized 4 µg MBP-tagged NUP214 FG-repeat peptide on amylose resin (New England Biolabs) together with NXF1–NXT1 (1:1 molar ratio) in buffer U (20 mM HEPES pH 7.9, 100 mM NaCl, 10% (w/v) glycerol, 0.1% (v/v) Igepal CA-630). Each binding reaction additionally contained equimolar amounts of THO, THO$^{\Delta THOC6}$, and THOC1/2/3 complex relative to NXF1–NXT1 and these were incubated together at 4°C on a rotating wheel. The beads were washed three times with 1 mL buffer U, and eluted for 1 hr on ice in buffer U containing 12 mM maltose. The elutions were visualized by SDS-PAGE (Coomassie blue). Each pulldown was repeated in triplicates.

## Negative stain electron microscopy of the endogenous THO–UAP56 complex

For negative stain EM imaging of the endogenous THO–UAP56 complex, copper grids were coated with a ~ 5 nm homemade carbon film and glow-discharged. 4 µL sample was applied to the grid and incubated for 1 min. The grid was blotted and washed four times with 4 µL distilled water, stained

for 1 min in 4 µL 2% (w/v) uranyl-acetate solution and blotted until dry. 1660 micrographs were acquired using SerialEM (*Mastronarde, 2005*) on a FEI Tecnai G$^2$ 20 transmission electron microscope (Eagle 4 k HS CCD camera) operated at 200 keV at a nominal magnification of 50,000x (2.21 Å pixel$^{-1}$) and a defocus range of –1 µm to –1.5 µm. 60938 particles were picked and extracted with a 256$^2$ pixel box size using WARP 1.07 (*Tegunov and Cramer, 2019*) and transferred to RELION 3.1 (*Scheres, 2012*) for 2D classification using default settings.

## Cryo-electron microscopy of the recombinant THO–UAP56 complex

4 µL concentrated and crosslinked THO–UAP56 complexes were applied to glow-discharged Cu R1.2/1.3 300 mesh holey carbon grids (Quantifoil). Grids were blotted at 4°C and 70% humidity and plunged into liquid ethane using a Leica EM GP. Cryo-EM data was recorded using SerialEM (*Mastronarde, 2005*) in two sessions (data sets 1 and 2) on a FEI Titan Krios G3i operated at 300 keV, equipped with Gatan K3 direct electron detector. Datasets 1 (7482 movies) and 2 (18821 movies) were acquired with a defocus range of –0.4 to –3.7 µm at a nominal magnification of 105,000x (0.86 Å pixel$^{-1}$). The camera was operated in 'super-resolution' mode (0.43 Å pixel$^{-1}$) with an exposure time of 3.8 s and 33 frames per micrograph, a dose rate of 9.73 e$^-$ pixel$^{-1}$ s$^{-1}$ and a total dose of 50 e$^-$ Å$^{-2}$.

## Image processing

Movie alignment with 4 × 6 patches, dose-weighting, and contrast transfer function (CTF) parameter estimation were all carried out in WARP 1.07 (*Tegunov and Cramer, 2019*) for the separate datasets 1 and 2. Automated particle picking was performed with a re-trained neural network in WARP 1.07 (*Tegunov and Cramer, 2019*), yielding 414,082 and 1,156,183 particles for datasets 1 and 2, respectively. The particles were then extracted, normalized, and Fourier cropped to 1.34 Å pixel$^{-1}$ with a 440$^2$ pixel box size in RELION 3.1 (*Scheres, 2012*). The gold-standard Fourier shell correlation (FSC) (0.143 criterion) was used to determine resolution, and B-factors were estimated and applied in RELION 3.1 (*Scheres, 2012*).

The initial 3D model of the THO–UAP56 complex was determined from the first 30,000 particles from dataset one with an ab initio refinement in cryoSPARC 2.0 (*Punjani et al., 2017*) using default parameters and two classes. The resultant class one was filtered to 60 Å and used as reference for separate 3D refinements in RELION 3.1 (*Scheres, 2012*) of all particles from each dataset, in order to align all particle images to a common reference. To increase data set size, we separately extracted two THO–UAP56 tetramer units from each octamer using the particle extraction parameters as above. This was possible since the octamer apparently does not show particle orientations with two tetramers on top of each other (*Figure 1—figure supplement 1c*), yielding 828,164 and 2,312,366 tetramer units for datasets 1 and 2, respectively. Subsequent 3D classification was carried out without image alignment to identify homogenous particle groups (*Figure 1—figure supplement 2*). To identify high-quality tetramer units we classified each dataset in 3D into eight classes using a soft-edged mask in the shape of the THO–UAP56 complex monomers 1A and 2B generated with the volume eraser in UCSF Chimera (*Pettersen et al., 2004*) and RELION 3.1 (*Scheres, 2012*). 3D class 8 (Round 1a) and class 7 (Round 1b) were selected for their excellent density quality, resulting in 195,098 high-quality tetramer units for subsequent processing, resulting in the final density maps A-E (*Figure 1—figure supplements 2–4*). First, we carried out a 3D refinement of the high-quality tetramer data set with the THO–UAP56 monomer 1A/2B mask, yielding a density (map B) with an overall resolution of 3.3 Å and a B-factor of −132 Å$^2$. To better resolve the connecting density to THO–UAP56 monomers 1B and 2A, we prepared a soft mask enveloping the complete tetramer, yielding a refinement from the same particles to 3.9 Å and a B-factor of −176 Å$^2$ (map A). THO–UAP56 monomer 1A THOC2, −3, −5, −7 and UAP56 regions remained poorly resolved, and were subjected to a focused refinement with a soft-edged mask surrounding these parts, to a resolution of 4.6 Å and a B-factor of −235 Å$^2$ (map D). THO–UAP56 monomer 1B THOC5 and THOC6 as well as the four-helix bundle connecting to monomers 1A and 1B were improved in a focused refinement of this region to resolution of 4.7 Å and a B-factor of −244 Å$^2$ (map E). To locally improve the density of THO–UAP56 monomer 2B THOC2–THOC3–UAP56, we performed an additional round of 3D classification (Round 2) yielding a subset of particles that could be subsequently refined to a resolution of 4.7 Å and a B-factor of −160 Å$^2$ (map C).

We used ResMap (*Kucukelbir et al., 2014*) to estimate the local resolution (*Figure 1—figure supplement 3c*) and performed 3D variability analysis of the high-quality tetramer units (6.2% of the total data) in cryoSPARC 2.0 (*Punjani et al., 2017*; *Figure 4—figure supplement 1d*).

## Structural modeling

We prepared a composite THO–UAP56 complex model by combining the cryo-EM densities A-E (*Figure 1—figure supplements 1d*, *2* and *3*). The structure was manually built in COOT (*Emsley and Cowtan, 2004*) using THOC2 MIF4G domain and THOC3 homology models and the previously determined crystal structure of the UAP56 RecA2 lobe (*Shi et al., 2004*). The model coordinates were refined into the respective sharpened maps B and D in PHENIX (*Adams et al., 2010*) using the phenix.real_space_refine routine, applying secondary structure and rotamer restraints. We first built monomers 1A and 2B. THOC1$_{2B}$ (residues 10–392), THOC2$_{2B}$ (residues 164–288), THOC5$_{2B}$ (residues 47–227), THOC7$_{2B}$ (residues 22–181), THOC5$_{1A}$ (tRWD), and THOC6$_{1A}$ were modeled into map B. The THOC2$_{2B}$ anchor helices were putatively assigned and modeled as poly-alanine with unknown register. To build the THOC2$_{1A}$ MIF4G domain (residues 535–687) we first prepared a homology model of the CWC22 MIF4G-like domain crystal structure (*Buchwald et al., 2013*) (PDB ID 4C9B), and then fitted this into map D and manually adjusted and extended the model to THOC2$_{1A}$ residue 288. The THOC2$_{1A}$ C-terminal residues 688–1175 (stern) were assigned based on density connectivity in map D, and were modeled as poly-alanine owing to a lower local resolution of ~5–6 Å (*Figure 1—figure supplement 3c*). Note that the connectivity between THOC2$_{1A}$ residues 688–895 remains uncertain. The THOC3$_{1A}$ homology model was generated with MODELLER (*Webb and Sali, 2017*) based on the WDR61 crystal structure (*Xu and Min, 2011*) (PDB ID 3OW8), rigid-body fitted into map D, and refined in real space using the Relax protocol in ROSETTA (*Tyka et al., 2011*). The human UAP56$_{1A}$ RecA2 structure (*Shi et al., 2004*) (PDB ID 1XTJ) was rigid-body fitted into map D and the interface with THOC2 was adjusted. To extend monomer 2B, THOC2$_{1A}$ (residues 240–1175)–THOC3$_{1A}$–UAP56$_{1A}$ were superimposed on THOC2$_{2B}$ as a rigid body, and this fit is in agreement with map C. To extend the monomer 1A, refined THOC1$_{2B}$, THOC2$_{2B}$ (residues 164–287), THOC5$_{2B}$ (residues 47–227), and THOC7$_{2B}$ were fitted into map D as rigid bodies. Further, THOC5$_{1A}$ (tRWD), and THOC6$_{1A}$ were rigid body fitted into map B, into the density of monomer 1B, and adjusted at the THOC5$_{1A}$–THOC5$_{1B}$ tRWD domain interface. To generate the complete THO–UAP56 complex model the refined monomer 1A, monomer 1B THOC5 tRWD and THOC6, and monomer 2B models were fitted into map A into the symmetry related positions of monomers 2A, 2B, and 1B, respectively, using COOT (*Emsley and Cowtan, 2004*). The THOC5$_{1A/1B}$–THOC7$_{1A/1B}$ four-helix bundle was modeled as poly-alanine into maps A and E and refined, and then copied as a rigid body into the THOC5$_{2A/2B}$–THOC7$_{2A/2B}$ position. ADP refinement was carried out in a composite THO–UAP56 map, generated from the individual maps A-E using phenix.combine_focused_maps (*Adams et al., 2010*). The final model comprises 28 proteins.

Based on our THO–UAP56 model, we revised the 6.0 Å resolution model of the chimeric yeast THO–Sub2 (PDB ID 5SUQ) crystal structure (*Figure 4—figure supplement 3a*, *Supplementary file 1*). We first defined a putative THO–Sub2 monomer A model that comprised residues 8–5289 (chain M) from the crystal structure. We then superimposed this monomer A model onto residues 6170–8550 (chain M) in COOT (*Emsley and Cowtan, 2004*), showing an excellent fit to the electron density and revealing the presence of a second monomer B. Four pieces of additional evidence validate this re-interpretation of the yeast crystal structure. First, placement of monomer A into the monomer B position could explain unmodeled density (*Figure 4—figure supplement 3b*). Second, the positions of phosphotungsten are consistent with a dimer architecture: M8701 and M8702 are now bound to equivalent positions in monomer A and B; N8701 binds the monomer B Sub2 RecA2 lobe in an equivalent position to A501 (*Figure 4—figure supplement 3b*). Third, weak density is observed in the location of the second Sub2 RecA2 lobe. Fourth, completion of the yeast THO-Sub2 dimer model with the monomer B Tho2 Stern, Tex1 and Sub2 fits into the asymmetric unit (*Figure 4—figure supplement 3c*) and is compatible with the observed crystal packing. The Tho2 C-terminus, corresponding to the human THOC2 Stern, is not resolved in monomer B. This might be due to the flexibility observed between Tho2 MIF4G and Stern domains (joint 2, *Figure 4—figure supplement 3d*) and the lack of stabilizing crystal packing contacts that are present in monomer A. We also see no density for monomer B Tex1, however, based on the 1:1 stoichiometry of Tex1 (or THOC3) in yeast (*Ren et al., 2017*) and human THO complexes (*Figure 1*, *Figure 1—figure*

*supplement 1*), its presence is likely. Overall, minor differences remain between the yeast and human THO–UAP56 monomer models that may stem from species differences and/or the unique THO–Sub2 architecture captured in the crystal (*Ren et al., 2017*).

Figures were made with UCSF Chimera X (*Goddard et al., 2018*) and PyMol (https://www.pymol.org).

## Acknowledgements

We thank the Plaschka group for their help and discussions; the Protein Technologies facility at the Vienna BioCenter Core Facilities GmbH (VBCF), a member of the Vienna BioCenter (VBC), for assistance with protein production; the VBCF Electron Microscopy Facility, in particular T Heuser and H Kotisch, for support and maintaining facilities; V-V Hodirnau at Institute of Science and Technology Austria EM facility for cryo-EM data collection; J Zuber and M Hinterndorfer for help with human cell culture and cloning; R Zimmermann and his team for computation support; M Novatchkova for bioinformatics analysis; K Mechtler and his team for mass spectrometry; the in-house Molecular Biology Service for reagents; and C Bernecky, A Pauli, R Pavri, and A Stark for discussions. We also thank C Bernecky, M Elmaghraby, D Haselbach, A Pauli, G Riddihough (Life Science Editors), A Stark, M Vorländer, and members of the Plaschka group for critical reading of the manuscript. We thank BP-F for drawing *Figure 4c*. UH was supported by JB and an SNF Early Postdoc Mobility fellowship (P2GEP3_188343). JB was supported by the Austrian Academy of Sciences, the European Research Council (ERC-2015-CoG 682181), and the Austrian Science Fund (F4303 and W1207). CP was supported by Boehringer Ingelheim and the European Research Council (ERC-2020-STG 949081 RNApaxport).

## Additional information

### Funding

| Funder | Grant reference number | Author |
| --- | --- | --- |
| Schweizerischer Nationalfonds zur Förderung der Wissenschaftlichen Forschung | P2GEP3_188343 | Ulrich Hohmann |
| H2020 European Research Council | ERC-2015-CoG 682181 | Julius Brennecke |
| Austrian Science Fund | F4303 and W1207 | Julius Brennecke |
| Österreichischen Akademie der Wissenschaften | | Julius Brennecke |
| Austrian Academy of Sciences | | Julius Brennecke |
| Boehringer Ingelheim | | Clemens Plaschka |
| H2020 European Research Council | ERC-2020-STG 949081 | Clemens Plaschka |

The funders had no role in study design, data collection and interpretation, or the decision to submit the work for publication.

### Author contributions

Thomas Pühringer, Ulrich Hohmann, Formal analysis, Investigation, Visualization, Writing - review and editing; Laura Fin, Ulla Schellhaas, Investigation; Belén Pacheco-Fiallos, Investigation, Visualization; Julius Brennecke, Supervision, Funding acquisition, Writing - review and editing; Clemens Plaschka, Conceptualization, Formal analysis, Supervision, Funding acquisition, Investigation, Visualization, Writing - original draft, Project administration, Writing - review and editing

### Author ORCIDs

Thomas Pühringer [ORCID] https://orcid.org/0000-0001-9127-9120
Ulrich Hohmann [ORCID] https://orcid.org/0000-0003-2124-1439

Ulla Schellhaas [iD] http://orcid.org/0000-0002-9684-9839
Clemens Plaschka [iD] https://orcid.org/0000-0002-6020-9514

**Decision letter and Author response**
Decision letter https://doi.org/10.7554/eLife.61503.sa1
Author response https://doi.org/10.7554/eLife.61503.sa2

## Additional files

### Supplementary files
• Supplementary file 1. Coordinate file of the revised yeast THO–Sub2 complex.

• Transparent reporting form

### Data availability
Three-dimensional cryo-EM density maps A, B, C, D, and E have been deposited in the Electron Microscopy Data Bank under the accession numbers EMD-11853, EMD-11857, EMD-11854, EMD-11855, EMD-11856, respectively. The coordinate file of the human THO-UAP56 complex has been deposited in the Protein Data Bank under the accession number 7APK.

The following datasets were generated:

| Author(s) | Year | Dataset title | Dataset URL | Database and Identifier |
|---|---|---|---|---|
| Pühringer T, Hohmann U, Fin L, Pacheco-Fiallos B, Schellhaas U, Brennecke J, Plaschka C | 2020 | human THO-UAP56 complex | https://www.rcsb.org/structure/7APK | RCSB Protein Data Bank, 7APK |
| Pühringer T, Hohmann U, Fin L, Pacheco-Fiallos B, Schellhaas U, Brennecke J, Plaschka C | 2020 | Three-dimensional cryo-EM density map A | https://www.ebi.ac.uk/pdbe/entry/emdb/EMD-EMD-11853 | Electron Microscopy Data Bank, EMD-11853 |
| Pühringer T, Hohmann U, Fin L, Pacheco-Fiallos B, Schellhaas U, Brennecke J, Plaschka C | 2020 | Three-dimensional cryo-EM density map B | https://www.ebi.ac.uk/pdbe/entry/emdb/EMD-EMD-11857 | Electron Microscopy Data Bank, EMD-11857 |
| Pühringer T, Hohmann U, Fin L, Pacheco-Fiallos B, Schellhaas U, Brennecke J, Plaschka C | 2020 | Three-dimensional cryo-EM density map C | https://www.ebi.ac.uk/pdbe/entry/emdb/EMD-EMD-11854 | Electron Microscopy Data Bank, EMD-11854 |
| Pühringer T, Hohmann U, Fin L, Pacheco-Fiallos B, Schellhaas U, Brennecke J, Plaschka C | 2020 | Three-dimensional cryo-EM density maps D | https://www.ebi.ac.uk/pdbe/entry/emdb/EMD-EMD-11855 | Electron Microscopy Data Bank, EMD-11855 |
| Pühringer T, Hohmann U, Fin L, Pacheco-Fiallos B, Schellhaas U, Brennecke J, Plaschka C | 2020 | Three-dimensional cryo-EM density maps E | https://www.ebi.ac.uk/pdbe/entry/emdb/EMD-EMD-11856 | Electron Microscopy Data Bank, EMD-11856 |

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
