## [Decision Letter]

**Acceptance summary:**

The manuscript describes the structure of the human THO-UAP56 complex determined by cryo-EM analysis. This is the first high-resolution structure of this large complex and this study will be of significance to the mRNP biogenesis and "R loop" fields providing an important basis for future mechanistic studies.

**Decision letter after peer review:**

Thank you for submitting your article "Human THO-UAP56 complex structure reveals multivalent interactions aid mRNA nuclear export" for consideration by *eLife*. Your article has been reviewed by two peer reviewers, and the evaluation has been overseen by a Reviewing Editor and James Manley as the Senior Editor. The reviewers have opted to remain anonymous.

The reviewers have discussed the reviews with one another and the Reviewing Editor has drafted this decision to help you prepare a revised submission.

Summary:

The manuscript entitled, "Human THO-UAP56 complex structure reveals multivalent interactions aid mRNA nuclear export", by Pühringer et al. describes the structure of the human THO-UAP56 complex determined by cryo-EM analysis. The six human THO proteins and the UAP56 helicase build up a 28-subunit tetramer assembly consisting of two asymmetric dimers. Modeling the export adapter ALY/REF into the structure suggests that a single ALY/REF protein bridges two proximal UAP56 helicases. The authors do not present biochemical experiments, e.g., using structure-based mutants, to underline the functional significance of the determined structure or test the validity of the proposed mechanism. However, since this is the first high-resolution structure of the large THO-UAP56 complex this study will be of significance to the mRNP biogenesis and "R loop" fields and will provide an important basis for new mechanistic studies.

Essential revisions:

1) The role of CHTOP

An interesting observation from this work is that THO-Sub2 from yeast is dimeric whereas human THO-UAP56 is a tetramer. Whilst the present structure explains this difference in molecular terms it doesn't really address the purpose of the tetramerisation. Did the authors consider the action of the mRNA export co-adaptor and TREX subunit CHTOP in the human mRNA export process? CHTOP binds the central NTF2L domain of NXF1 and together with ALYREF drives NXF1 into an open conformation which allows it to bind RNA (https://pubmed.ncbi.nlm.nih.gov/22893130/) and (https://pubmed.ncbi.nlm.nih.gov/23299939/) and (https://pubmed.ncbi.nlm.nih.gov/31104896/). Importantly, CHTOP has tandem UAP56 binding motifs separated by only a few amino acids so if it were to bind two UAP56 molecules they would presumably need to be very close, as two of them are in this structure. Are the UAP56 molecules close enough though to simultaneously bind the CHTOP tandem UBMs? The authors should look into this. CHTOP also activates the helicase and ATPase activities of UAP56 just like ALYREF. Efficient loading of NXF1 onto the mRNP needs both ALYREF and CHTOP, so this tetrameric structure might provide that key platform allowing one UAP56 juxtaposed dimer to deliver ALYREF to RNA and the other to deliver CHTOP.

In yeast, the central NTF2L domain has an unusual insertion which allows it to directly bind RNA (https://pubmed.ncbi.nlm.nih.gov/17434126/) not present in human NXF1 and there are no reports of co-adaptor like activities in yeast binding this NTF2L domain so perhaps THO-Sub2 is dimeric because it doesn't need to recruit a CHTOP like molecule, just Yra1p? The authors should consider these ideas in their revision and certainly explore how CHTOP might bind UAP56 in their structure given its tightly juxtaposed tandem arrangement of UBMs.

2) The title is grammatically awkward and should be changed.

3) It would be easier to understand the structure of the complex, if the monomer was discussed first, followed by a description of the tetramer.

---

## [Author Response]

Essential revisions:1) The role of CHTOPAn interesting observation from this work is that THO-Sub2 from yeast is dimeric whereas human THO-UAP56 is a tetramer. Whilst the present structure explains this difference in molecular terms it doesn't really address the purpose of the tetramerisation. Did the authors consider the action of the mRNA export co-adaptor and TREX subunit CHTOP in the human mRNA export process? CHTOP binds the central NTF2L domain of NXF1 and together with ALYREF drives NXF1 into an open conformation which allows it to bind RNA (https://pubmed.ncbi.nlm.nih.gov/22893130/) and (https://pubmed.ncbi.nlm.nih.gov/23299939/) and (https://pubmed.ncbi.nlm.nih.gov/31104896/). Importantly, CHTOP has tandem UAP56 binding motifs separated by only a few amino acids so if it were to bind two UAP56 molecules they would presumably need to be very close, as two of them are in this structure. Are the UAP56 molecules close enough though to simultaneously bind the CHTOP tandem UBMs? The authors should look into this. CHTOP also activates the helicase and ATPase activities of UAP56 just like ALYREF. Efficient loading of NXF1 onto the mRNP needs both ALYREF and CHTOP, so this tetrameric structure might provide that key platform allowing one UAP56 juxtaposed dimer to deliver ALYREF to RNA and the other to deliver CHTOP.In yeast, the central NTF2L domain has an unusual insertion which allows it to directly bind RNA (https://pubmed.ncbi.nlm.nih.gov/17434126/) not present in human NXF1 and there are no reports of co-adaptor like activities in yeast binding this NTF2L domain so perhaps THO-Sub2 is dimeric because it doesn't need to recruit a CHTOP like molecule, just Yra1p? The authors should consider these ideas in their revision and certainly explore how CHTOP might bind UAP56 in their structure given its tightly juxtaposed tandem arrangement of UBMs.

We thank the reviewers for their suggestions and hypothesis regarding CHTOP, and apologize for its omission in the earlier text. We now mention CHTOP and the additional UBM-containing export adapters in the Introduction and expanded their discussion in the subsection “TREX complex and ALYREF function”. We wish to stress that in this study we primarily focused on ALYREF due to its high conservation from yeast to humans, ubiquitous expression in human tissues, and the available structural and biochemical data from yeast and human systems.

The tandem UBMs in CHTOP are very interesting, but we think that they are very unlikely to bridge two UAP56 helicases within the THO–UAP56 tetramer (now also included in Figure 4—figure supplement 2C as suggested below). A secondary structure prediction using PsiPred (Buchan and Jones, 2019) suggests that both CHTOP UBMs are part of one long helix. A comparison of the CHTOP UBMs to the Yra1 C-terminal UBM bound to Sub2 (PDB-ID: 5sup, Ren, Schmiege and Blobel, 2017) reveals a spacing of seven residues between the anchoring tyrosine of the first UBM (CHTOP Tyr223) and the first tightly bound residue of the second UBM (CHTOP Leu231). Due to the rotation of the two UBMs along the helical axis it may indeed be possible that two UAP56 molecules bind simultaneously to CHTOP, however, the spacing is too close to span the distance between the RecA1 lobes of two UAP56 molecules in the context of the THO–UAP56 tetramer. The THO–UAP56 tetramer architecture does, however, suggest that the multiplicity of UBM binding sites may increase the affinity of UBM-containing proteins for their target mRNA substrates. We summarized these ideas in the subsection “TREX complex and ALYREF function”.

As the reviewers point out, we were also very intrigued to find a tetrameric human and dimeric yeast THO architecture. The reason for this architectural difference remains unclear, though we speculate it may reflect the complexity in gene architecture in complex eukaryotes. For the reasons below, we do not think that the dimer/tetramer organisation reflects a dependence on other export adapters, like CHTOP, however, the tetramer may have allowed for their emergence. First, the loop in the NTF2L domain, residues 407 and 436, in *S. cerevisiae* Mex67p is present only in *S. cerevisiae*, but absent in *S. pombe*, worms, plants, insects and vertebrates. In contrast, the 6-subunit tetrameric THO assembly is present in insects, worms, and plants but not in *S. pombe* (which lacks the THOC5 tRWD domain as well as THOC6, resulting in a dimeric THO complex). Second, CHTOP is not conserved outside of vertebrates. Thus, the presence of a tetrameric THO complex and CHTOP as well as the absence of the *S. cerevisiae*-specific loop in Mex67 do not seem to be related. We hope that knowledge of the tetrameric THO complex will inform further experiments to study the function of CHTOP. Within our group we are excited to address the potential roles of further export adapters and, in particular, the recruitment of NXF1 to the THO complex in future studies.

2) The title is grammatically awkward and should be changed.

We have changed the title in our revised manuscript. It now reads: “Structure of the human core transcription-export complex reveals a hub for multivalent interactions”. We reason that since all TREX complexes are thought to contain THO and UAP56, but as the reviewers point out, not necessarily ALYREF, we can consider THO–UAP56 the TREX core.

3) It would be easier to understand the structure of the complex, if the monomer was discussed first, followed by a description of the tetramer.

We thank the reviewers for this suggestion. While preparing the current manuscript we discussed and tested various orders of describing the THO-UAP complex structure. We believe this to be the more accessible route (tetramer to monomer), but we appreciate that parts of the structural description in the subsection “THO–UAP56 structure” were not easy to follow. To address this, we edited several key parts in the revised version to improve readability (subsection “THO–UAP56 structure”). In the new version, we first briefly describe the monomer, before delving into the dimer/tetramer organisation, and then returning to the monomer. Starting out with the detailed dimer/tetramer organization allows us to later focus fully on the monomer, the more conserved THOC1, -2, -3, UAP56 subunits, NXF1, and ALYREF, and we hope the reviewers agree with this revised version.